# Characterization of Engineered Cerium Oxide Nanoparticles and Their Effects on Lung and Macrophage Cells

Michael Bushell [1], Filip Kunc [1], Xiaomei Du [2], Andre Zborowski [2], Linda J. Johnston [1]  and David C. Kennedy [1,*]

[1] Metrology, National Research Council Canada, 1200 Montreal Road, Ottawa, ON K1A 0R6, Canada
[2] Energy, Mining and Environment, National Research Council Canada, 1200 Montreal Road, Ottawa, ON K1A 0R6, Canada
* Correspondence: david.kennedy@nrc-cnrc.gc.ca

**Abstract:** Cerium oxide nanoparticles are promising materials as novel nanoscale therapeutics and are commonly used materials in industrial processes. Most cytotoxicity studies on cerium oxide nanoparticles are made from in-lab prepared materials making comparison between studies challenging, especially when performed on unique cell lines under non-standard conditions. Using commercially available nanoparticles we show that particle stability/agglomeration may be critical in determining the cytotoxicity in some cell lines, while in other cell lines, larger sized primary particles are linked to higher cytotoxicity, contrasting what has been reported in the literature for smaller cerium nanoparticles. To accelerate the development of cerium oxide enabled commercial processes and biomedical innovations, a clearer understanding of the interactions between cerium oxide nanoparticles and cells is needed to better understand their fate in and impact on biological systems.

**Keywords:** cerium oxide; nanoparticles; toxicity; characterization; standards

## 1. Introduction

Cerium oxide nanoparticles are widely used industrial materials as catalysts in fuel combustion systems and as electrolytes in fuel cells [1,2]. They have also been used in glass polishing [3] and now are being investigated for a number of biological applications due to their antioxidant properties [4–6]. Cerium oxide nanoparticles have recently been investigated for activity against cancer tumours [7], inflammation [8], as antimicrobial agents [9–12], as therapeutics for neurodegenerative disorders such as Parkinson's disease [13] and Alzheimer's disease [14], and even for activity against COVID-19 [15,16]. Many of the studies on the biological impact of cerium oxide nanoparticles are focused on specifically designed particles with unique surface chemistries or targeting molecules for disease intervention. For the development of safe exposure levels and regulations for cerium oxide nanoparticles, there is a knowledge gap on the fate of industrially produced materials and their interaction with human cells. While several studies do report cytotoxicity of particles [17–25], the correlation between specific physical properties of a range of particles (size, shape, charge, surface area and surface chemistry) and exposure risk and cytotoxicity across different cell types or organisms is lacking. Nanotoxicology is plagued by several challenges including particle stability, differences in size and purity between samples, and the effects of changes in sample dispersion, pH, media composition and time, all of which can dramatically affect the measured outcomes. There are also challenges in comparing cell lines and even differences in cell culture medium composition that can affect particle dissolution rates and agglomeration, ultimately changing their bioavailability [26,27]. Here, we have compared the results of 8 different commercially available cerium oxide nanoparticles from three manufacturers, spanning different nanoscale sizes and with different surface coatings. These particles have been characterized extensively in order to correlate physical chemical properties when possible with measured cytotoxicity values in both lung epithelial (A549) and mouse macrophage (J774A.1) cells. Biological endpoints

and particle stability have been measured at various time points, and agglomeration and primary particle size in cell culture medium are discussed as determining factors for cellular cytotoxicity.

## 2. Experimental

### 2.1. Materials

Cerium oxide ($CeO_2$) nanoparticle samples were purchased as dry powders from US Research Nanomaterials (Ce-01, Ce-02, Ce-03, Ce-04, Ce-05) Nanostructured & Amorphous Materials Inc. (Katy, TX, USA) (Ce-06), and mkNano (Missisauga, Canada) (Ce-07, Ce-08).

### 2.2. Cerium Oxide Nanoparticle Dispersion Optimization

Dry cerium oxide was weighed and deionized water (Milli Q, 18.2 MΩ cm) was added in two steps to yield a 0.1% by mass suspension as per a reported dispersion protocol for cerium oxide nanoparticles [28]. In the first step, 0.1 mL of deionized water was added to the cerium oxide powder, the sample was then mixed with a glass rod, after which an additional 0.1 mL was added and mixed. Once a thick paste was obtained, the remaining deionized water was added and used to wash any remaining material on the glass rod into the sample. The samples were then vortexed for 5 s and sonicated to the optimal sonication energy using a 130 W ultrasonic processor (EW-04714-50, Cole-Parmer) equipped with a ¼ inch tip probe (EW-04712-14 Cole-Parmer) and operated at 50% amplitude with 30 s on/off cycles. The sonicator probe was polished every 4th sample or after a total of 12,000 J delivered. The total energy transfer efficiency for the sonicator used is 0.97, as measured calorimetrically. To prevent overheating, the sample was immersed in a water-ice bath during the sonication cycle. The optimal sonication energy was determined for each sample by monitoring the particle size as assessed by dynamic light scattering (DLS) as a function of applied sonication energy; an example of the optimization for Ce-03 is shown in Figure S1. A table with all the DLS measurements for suspensions prepared with the optimized sonication energy can be found in the supplementary material (Table S1). Due to the stearic acid coating on Ce-05 ethanol was used as the dispersant and diluent instead of deionized water, although the dispersions obtained were still very aggregated (Figure S2); although the experiments to test for the optimal sonication energy were done in ethanol, all biological assays were dispersed in water and diluted in cell culture media as for the other samples. Zeta potential values for the suspensions as prepared in deionized water are provided in Table S1.

For biological testing, cerium oxide nanoparticles were suspended in water as reported above, and a stock suspension was prepared by diluting one-part aqueous nanoparticle suspension with one part complete Dulbecco's modified Eagle's medium (DMEM, Gibco (Oakville, Canada)) supplemented with 10% fetal bovine serum (FBS, Gibco) and 1% penicillin-streptomycin (pen/strep, 50 µg/mL, Gibco). This step effectively diluted the media by 50%, and to account for serum concentration, all subsequent dilutions were made with a 1:1 mixture of media and deionized water. This ensures that all experiments have the same serum concentration, as this will affect the toxicity of both particles and metal ions. 100 µL of the nanoparticle suspensions were then added to each well that already contained 100 µL of medium and cells, so the final serum concentration of all experimental wells was always 7.5% (100 µL of 10% and 100 uL of 5% mixed together).

### 2.3. Dynamic Light Scattering (DLS)

Samples for DLS measurements were prepared by diluting the 0.1% by mass metal oxide suspensions to 0.01% by mass using deionized water. The suspensions were analyzed with a Zetasizer Nano ZS (red) (Malvern) using a 632.8 nm HeNe laser and signal detection at 173°C. The measurements of Z-average (equivalent hydrodynamic diameter) and poly-dispersity index (PDI) were done at 25 °C using corrected values for viscosity and refractive index for the dispersants. Each sample had an initial temperature equilibration time of 180 s and 3 trials, each consisting of ten measurements of 10 s, were acquired. Each measurement

was analyzed using Zetasizer software (Malvern, ver. 7.11, Westborough, MA, USA) by the cumulants method with the general purpose model. The data were processed to obtain 3-measurement average values and corresponding standard deviations for Z-average and PDI. Several concentrations were measured (0.1–0.001%) initially to ensure that 0.01% by mass dispersions was a suitable concentration for DLS measurements. DLS measurements were taken within 10 min of sample sonication.

For biological samples, 100 µL cerium oxide nanoparticles suspended in water were added to 500 µL DMEM biological media (as described below) and 400 µL water to achieve the same 7.5% serum concentration as was used in the cell culture experiments, and a final particle concentration of 100 µg/mL, the same as measurements made in water alone. Samples were measured immediately upon preparation and then every 24 h for 3 days. Samples were incubated at 37 °C in the incubator during the entire time period to mimic conditions in the biological experiments. Samples were re-suspended (pipetted up and down until settled particles were resuspended) before measuring, as most of the particles for most samples had settled out of suspension during the interim 24 h incubations.

### 2.4. Transmission Electron Microscopy (TEM)

All samples were deposited on plasma treated carbon film covered copper grids (200 mesh, Ted Pella 01840-F). The carbon grids were treated with a Fischione 1070 NanoClean plasma treater, using a 75/25 $Ar/O_2$ mix at a flow rate of 30 standard $cm^3$/min at about 40 W for 2 min. Immediately after plasma treatment, 10 µL of 0.1 mg/mL or 0.01 mg/mL of the cerium oxide dispersion was added to the treated carbon film, and wicked away after 10 min using a piece of filter paper. The sample was then quickly immersed in deionized water and allowed to fully dry, typically for 2 h. For Ce-05 (stearic acid coated), the ethanol dispersion was allowed to fully dry without a wicking or washing step. Although samples were prepared at two concentrations (0.1 mg/mL, 0.01 mg/mL), the higher concentration (0.1 mg/mL) was only used for imaging if the 0.01 mg/mL sample gave TEM images with poor contrast or heavily aggregated particles.

TEM images were recorded on a Titan3 (80–300 kV) FEI microscope operated at 300 kV and calibrated with a TEM magnification standard (MAG*I*CAL, EMS). Images were analyzed with ImageJ using the polygon outlining feature to trace individual particles and record the particle area, perimeter, Feret and minFeret. Area was converted to equivalent circular diameter and aspect ratio was calculated as the Feret/minFeret ratio. Particle size histograms and smooth kernel probability distributions were plotted and statistics were calculated in OriginPro 2019.

### 2.5. Thermogravimetric Analysis (TGA)

Experiments were conducted using either a NETZSCH Iris TG209 F1 or a NETZSCH Jupiter STA449 F1 instrument coupled with a Bruker Tensor 27 FTIR spectrometer. In a typical experiment, 20–40 mg of a powdered cerium oxide sample was loaded to an empty aluminum oxide crucible which was previously annealed in a natural gas flame for approximately 30 s. The mass of the sample was adjusted to ensure that a total mass loss of at least 1 mg was obtained. The sample was inserted into the instrument under 50 mL/min Ar atmosphere (argon protective 25 mL/min) and left to stabilize for 1 h; the transfer line to the FTIR spectrometer was also purged with 50 mL/min of Ar. The thermal cycle 25–950 °C (10 °C/min, unless otherwise specified) was then initiated maintaining the Ar flow. All TGA experiments were run against the correction for an empty aluminum oxide crucible with the same argon atmosphere. Thermograms of bare metal oxides from the same supplier with the same reported size were run for comparison and to assist in identification of any mass loss that was due to functional groups. Mass loss temperatures are reported as the maxima in the derivative curve of the mass loss plot.

### 2.6. Specific Surface Area (SSA) Determination

The Brunauer Emmett Teller (BET) method with nitrogen adsorption was used for measurement of specific surface area (SSA) with an ASAP 2020 system from Micromeritics. The cell containing the sample was weighed before degassing. The samples were heated at 10 °C /min to 110 °C, held for 10 min and then heated to 200 °C at 10 °C/min and held for 2 h. Specific surface area was determined by the multipoint BET method.

### 2.7. Cell Culture

A549 and J774A.1 cells (American Tissue Culture Centre) were grown in Dulbecco's modified Eagle's medium (DMEM) (Gibco) supplemented with 10% fetal bovine serum (FBS) (Gibco) and 1% penicillin-streptomycin (Pen/strep) (50 µg/mL, Gibco) under standard culture conditions (37 °C, 5% $CO_2$). Cells were grown in T75 flasks (Falcon) and Trypsin-EDTA solution (Gibco) was used for passaging A549 cells (2 mL per T75 flask). For passaging J774A.1 cells, a cell scraper was used to detach cells from flask and the cells were then diluted one into ten in a new flask.

### 2.8. MTT Assay

Cells were seeded into wells in a 96-well plate (Falcon) ($1 \times 10^5$ cells/mL, 100 µL per well) to cover an $8 \times 6$ grid, filling 48 wells. Remaining wells were filled with 200 µL of PBS. After 24 h, 100 µL volumes of dilutions of particles in complete media spanning from 500 µg/mL to 5 µg/mL were added to the seeded wells (final concentrations spanning 250 µg/mL to 2.5 µg/mL). For each nanoparticle, seven dilutions were prepared and for each dilution three replicates were performed. In the remaining 6 wells, 100 µL of media was added as a particle-free control. Cells were then incubated with nanoparticles for 24, 48 and 72 h. For each time point, workup consisted of adding a 50 µL PBS solution of MTT (3-(4,5-dimethylthiazol-2-yl)-2,5-diphenyltetrazolium bromide, Sigma Aldrich (Oakville, Canada), 2 mg/mL) to each well and then incubating for 3 h. After 3 h, media was aspirated from all wells, leaving purple formazan crystals in those wells with viable cells. To each well, 150 µL of DMSO was added and plates were agitated manually to dissolve the crystals. 100 uL of each well was then transferred to a fresh plate. This was done in order to remove scattering from precipitated particles that affected the absorbance readings. Plates were then analyzed using a plate reader (Fluorstar Omega, BMG Labtech, Ortenberg, Germany) to determine the absorbance of each well at 570 nm. This reading divided by the average from the reading of the six control wells was plotted to determine the $IC_{50}$ value of each particle for each cell line. Three replicates were performed for each sample on each cell line at each time point.

### 2.9. Lactate Dehydrogenase (LDH) Assay

Kits were purchased from abcam and the assay adapted to align with the 72 h timescale we measured for other assays. Following 72 h treatment, 3 wells from the untreated cells were treated with cell lysis solution to create a positive control for LDH release, while the other 3 control wells served as the low end LDH baseline. 100 µL of media was drawn from each well and transferred to a new plate. 100 µL of reaction mixture was then added to each well and the plate left at room temperature for 30 min in the dark. Absorbance was then measured at 490 nm.

### 2.10. Dichlorodihydrofluorescein Diacetate (DCFDA) Assay

Cells were prepared in a manner identical as for the MTT assay but seeded into black walled 96-well plates and DSCFDA kits were purchased from abcam. Immediately prior to use, DCFDA buffer and solution was prepared as per the assay kit protocol. After seeding the cells overnight, the wells were washed with 100 µL DCFDA buffer. The wells were then filled with 100 µL DCFDA solution and incubated in standard culture conditions for 45 min. After incubation, the DCFDA solution was removed and replaced with 100 µL 1X PBS. The fluorescence was read at Ex/Em 485/535 using a spectrophotometer. The buffer

was then removed and replaced with 100 µL of cell culture media. Dilutions were then added in a manner identical as for the MTT assay and plates were scanned immediately to baseline the fluorescence in each well. Fluorescence measurements were taken again at Ex/Em = 485/535 nm after 1, 2, 3 and 4 h. Little effect was observed for all samples, so they were then incubated overnight and the fluorescence recorded again at 24 h. The plates were incubated under standard culture conditions between fluorescence measurements.

## 3. Results and Discussion

### 3.1. Nanoparticle Characterization

3.1.1. TEM Analysis

TEM images were obtained by depositing the optimized nanoparticle dispersions on TEM grids. Images were obtained at several resolutions since the particle size distributions were broad and there were differences between samples. Representative images are shown in Figure 1 for all cerium oxide nanoparticles; all samples showed irregularly shaped, aggregated particles and some samples had a broader range of particle sizes. Ce-03 (bare), Ce-04 (PVP coated) and Ce-05 (stearic acid coated) are all from the same supplier and have the same reported size but different coatings which might be expected to result in similar TEM contrast. Although all three samples were heavily aggregated, the aggregation coupled with poor contrast made it almost impossible to identify individual particles for size analysis for Ce-05. This may be caused by either a reduction in electron scattering of the particles due to a decrease in particle thickness or an increase in background scattering caused by contamination of the sample adsorbed onto the TEM grid. Since the three samples have similar sizes and are from the same supplier, it seems likely that the presence of stearic acid is at least partly responsible for the problems.

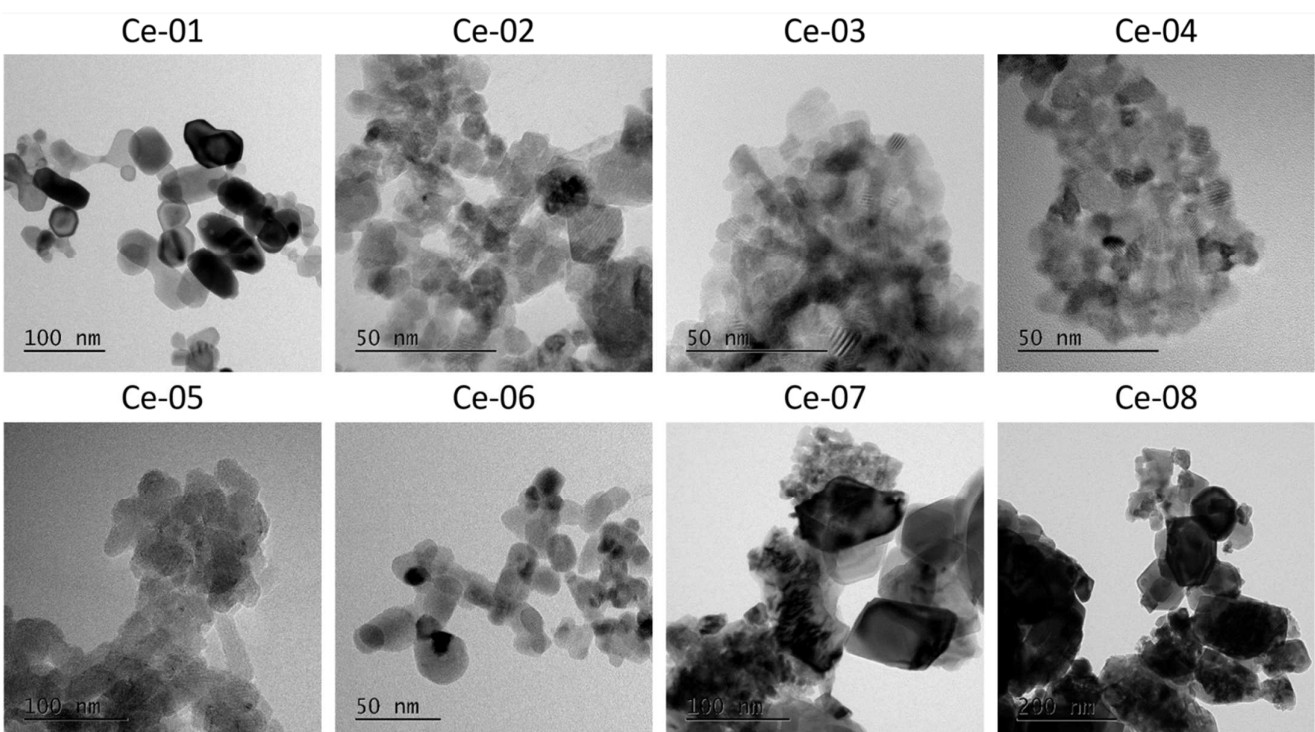

**Figure 1.** Representative TEM images for each cerium oxide sample.

TEM particle size analysis is summarized in Table 1. Particle size distributions are characterized by the mean equivalent circular diameter, with standard deviation as a measure of the distribution width; mean aspect ratio and standard deviation are also provided. It is important to note that a relatively small number of particles was analysed for three samples (Ce-02, Ce-03 and Ce-04) which means that the particle size distribution

is less well-defined. All but two nanomaterials (Ce-03 and Ce-04) had significantly (>15%) different equivalent circular diameters from the nominal size reported by the supplier. Notable examples are Ce-06 for which the measured diameter is more than a factor of 2 smaller than the supplier reported size and Ce-08 which has a measured diameter more than twice as large as the supplier reported size. It is interesting to note that Ce-07 and Ce-08 are from the same supplier and have different nominal sizes. However, their measured equivalent circular diameters are not significantly different as assessed by a one-way ANOVA test (95% confidence level). A Kolmogorov–Smirnov test also fails to identify a significant difference between the two data sets at the 95% confidence level, although it is possible that a significant difference might be detected with larger data sets.

**Table 1.** Particle size distributions for cerium oxide nanoparticles; data are summarized as equivalent circular diameters (mean, standard deviation, standard error and median values) and aspect ratios (mean and standard deviation). The nominal size provided by the manufacturer is also provided [1].

| Sample | Measurand | n [1] | Mean (nm) | Std [1] Dev (nm) | Std [1] Error | Median (nm) | Nominal Size (nm) |
|---|---|---|---|---|---|---|---|
| Ce-01, bare | Equiv diameter, nm | 152 | 27.4 | 10.8 | 0.9 | 25.8 | 50 |
| | Aspect ratio | | 1.31 | 0.21 | 0.02 | 1.25 | |
| Ce-02, bare | Equiv diameter, nm | 68 | 13.5 | 5.11 | 0.6 | 12.2 | 10–30 |
| | Aspect ratio | | 1.35 | 0.20 | 0.02 | 1.32 | |
| Ce-03 bare | Equiv diameter, nm | 55 | 9.5 | 2.0 | 0.3 | 9.0 | 10 |
| | Aspect ratio | | 1.37 | 0.20 | 0.03 | 1.33 | |
| Ce-04 PVP | Equiv diameter, nm | 88 | 10.3 | 2.2 | 0.2 | 10.5 | 10 |
| | Aspect ratio | | 1.35 | 0.21 | 0.02 | 1.31 | |
| Ce-05 [2] stearic acid | Equiv diameter, nm | | | | | | 10 |
| | Aspect ratio | | | | | | |
| Ce-06, bare | Equiv diameter, nm | 180 | 19.6 | 11.2 | 0.8 | 16.8 | 50–105 |
| | Aspect ratio | | 1.39 | 0.27 | 0.02 | 1.31 | |
| Ce-07, bare | Equiv diameter, nm | 103 | 57.6 | 24.8 | 2.4 | 52.3 | 70 |
| | Aspect ratio | | 1.40 | 0.24 | 0.02 | 1.36 | |
| Ce-08, bare | Equiv diameter, nm | 108 | 64.3 | 27.7 | 2.7 | 55.3 | 25 |
| | Aspect ratio | | 1.38 | 0.23 | 0.02 | 1.32 | |

[1] The number of particles analyzed is n; standard deviation (Std Dev) provides a measure of the breadth of the particle size distribution, the standard error (Std error) is for the mean and Equiv diameter represents the equivalent circular diameter. [2] Sample not analyzed.

Particle size distributions are represented in Figure 2 as kernel distribution plots. The kernel plots are smoothed histograms and provide a simple representation of the nanoparticle size distributions, allowing for qualitative comparisons between the samples in one figure. Most samples had relatively broad diameter distributions, with Ce-01, Ce-06, Ce-07 and Ce-08 having the largest equivalent diameter distributions (expressed as the ratio of distribution width (standard deviation)/mean equivalent diameter), Table 1 and Figure 2, right. Note that some samples appear to have multiple sub-populations of different sizes

(Ce-01, Ce-02, Ce-03 and Ce-04) although this is primarily due to the smoothing of the histogram to generate the kernel distribution plots and a much larger number of particles would be required to confirm the presence of sub-populations. It is important to note that the wide range of particle sizes (Ce-01, Ce-07 and Ce-08) may influence the particle's properties, and potential toxicity; for example, small particles may well exhibit different localization or cytotoxicity compared to larger particles. The aspect ratio kernel plots (Figure 2, Left) indicate that there are only small differences in the distribution mean and width, with Ce-06 having the widest distribution (expressed as a ratio of distribution width/mean diameter) and a larger fraction of larger particles. The results are consistent with the qualitative observations from the TEM images, none of which provide evidence for high aspect ratio rod-shaped particles. Although most nanoparticles are irregular in shape, the mean aspect ratios were all between 1.3 and 1.4 (Table 1).

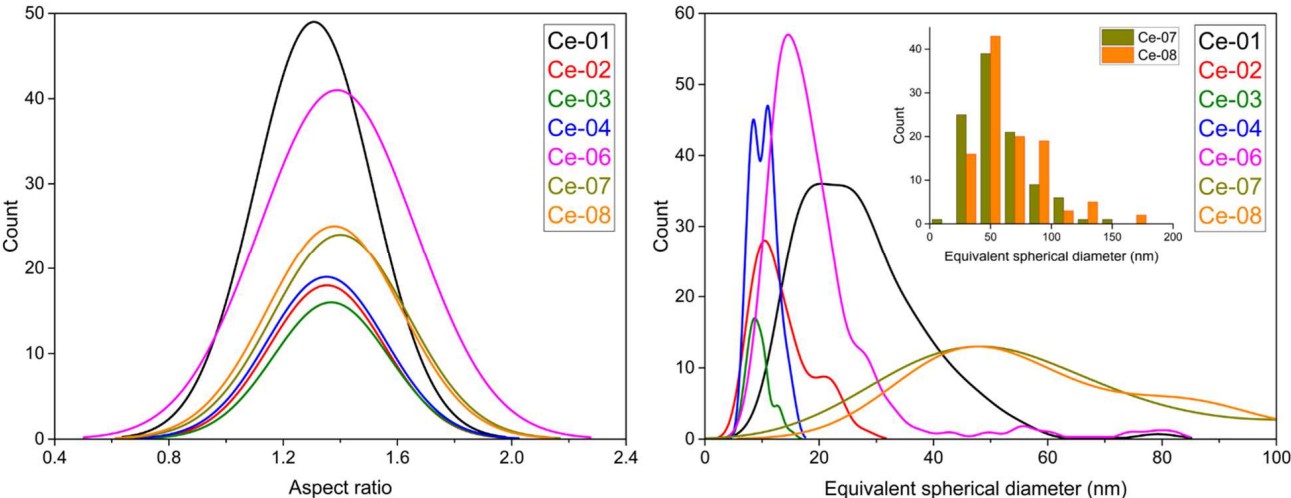

**Figure 2.** (Left) kernel smooth aspect ratio distributions. (Right) Kernel smooth equivalent spherical diameter distributions for the cerium oxide nanoparticles ($\geq$100 nm), with the inset showing the expanded histograms for Ce-07 and Ce-08.

### 3.1.2. BET Analysis

The Brunauer–Emmett–Teller method was used to determine the specific surface area (SSA) of the nanoparticle samples. The SSA measured by BET can indicate the size and level of aggregation of the particles within the solid material. High SSA values measured for smaller particles which possess larger surface to volume ratios, while lower values, are consistent with particles that are strongly aggregated, inhibiting nitrogen penetration. The experimentally determined SSA values and the reported values from the supplier can be found in Table 2. It should be noted that the supplier provides a range of SSA values for each sample, so one cannot make a direct comparison to the data for the specific samples used in this study. Three samples (Ce-02, Ce-05, Ce-06) had measured SSA values that were outside the range cited by the supplier, while the measured values for three other samples (Ce-01, Ce-03 and Ce-04) fell within the supplier range. The experimentally determined SSA values for Ce-03 (uncoated) and Ce-04 (PVP) were similar, with SSA value of 73.2 m$^2$g$^{-1}$ and 61.5 m$^2$g$^{-1}$, respectively. This is in contrast to the stearic acid coated sample (Ce-05) which has the same nominal particle size but a significantly smaller SSA of 26.9 m$^2$g$^{-1}$. This large decrease in SSA for Ce-05 is consistent with an increase in aggregation in the nanopowder, impeding nitrogen penetration and is in good agreement with the qualitative observations from TEM images and the measured Z-average from DLS (Figure S2 and Table S1). This large amount of aggregation was also observed in the dispersion from the DLS studies (Supplementary, Figure S2 and Table S1).

**Table 2.** Comparison of the experimental specific surface area (SSA) and the supplier reported values.

| Sample | Coating | SSA, Supplier (m²/g) | SSA, BET (m²/g) |
|--------|---------|----------------------|-----------------|
| Ce-01 | uncoated | 30–35 | 27.4 |
| Ce-02 | uncoated | 30–50 | 63.7 |
| Ce-03 | uncoated | 35–75 | 73.2 |
| Ce-04 | PVP | 35–75 | 61.5 |
| Ce-05 | Stearic acid | 35–75 | 26.9 |
| Ce-06 | uncoated | 8–15 | 25.8 |
| Ce-07 | uncoated | - | 12.5 |
| Ce-08 | uncoated | - | 12.7 |

In general, as a nanoparticle gets smaller its surface area to volume ratio increases, this can be measured as an increase in SSA. Figure 3 looks at this relationship, comparing the equivalent diameter measured by TEM to the experimentally determined SSA for each sample. The SSA was approximated for particle sizes between 10–100 nm, using the reported CeO$_2$ density of 7.132 g/cm³ from US Research Nanomaterials (Figure 3, black dashed line) and assuming perfectly spherical particles. Although this approximation ignores surface effects and shape variability for the nanoparticles, the overall trend matches well with the observed increase in SSA for particles with a smaller mean equivalent diameter, with the exception of Ce-06, for which there is a larger difference between the measured and calculated SSA.

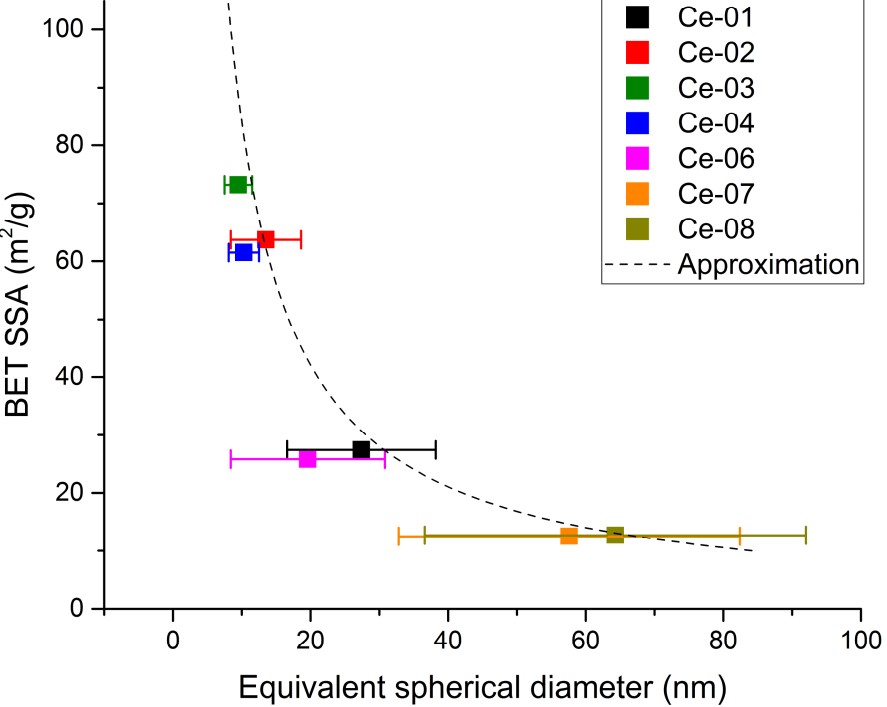

**Figure 3.** Correlation of the measured nanoparticle size with its SSA as measured by BET. The data points represent the TEM mean equivalent circular diameter and error bars are the standard deviations on the mean. The dashed line shows for the change in SSA as a function of particle size calculated using the US Research Nanomaterials reported CeO$_2$ density of 7.132 g/cm³ and assuming spherical particles.

### 3.1.3. Quantification of Surface Coatings TGA

The surface coating content for the PVP and stearic acid modified cerium oxide nanoparticles was estimated from the mass loss observed by TGA, as described previously for a series of metal oxides with different surface modifications [29,30]. This work demonstrated that TGA provides reasonably accurate quantification for small nanoparticles (e.g., 10–30 nm) with large surface areas, as assessed by benchmarking the TGA results against quantitative NMR analysis of coatings and functional groups extracted from the nanomaterial surface.

Representative TGA results are shown in Figure S3. The results under argon, when coupled with FTIR spectra of the evolved gases, provide information that can assist with identification of the surface coatings. A thermogram for the bare sample (Ce-03) indicates that the bare sample has mass loss around 100 °C that corresponds to water, but little mass loss at higher temperatures where organic coatings are expected. For Ce-04, a peak with a maximum at 430 °C can be assigned to PVP, consistent with the FT-IR spectrum of evolved gases in Figure S3d which shows a carbonyl signal at ~1770 cm$^{-1}$. A more complex pattern with maxima at 310 °C and 440 °C is obtained for stearic acid; this was previously suggested to reflect the presence of both acid and carboxylate binding on the surface [29]. Both coated samples showed mass loss above 600 °C which is not included in the estimation of the surface coating content. The data both with and without correction for mass loss for a bare sample of the same size from the same supplier is summarized in Table 3. Note that for both samples the coating content measured by TGA is 10–20% lower than the values obtained previously by NMR.

**Table 3.** Quantification of surface coatings using TGA [1].

| Sample | Coating | TGA in Argon, μmol/g, (n) [1] (Temperature Range) | |
| --- | --- | --- | --- |
| | | **Corrected** | **Uncorrected** |
| Ce-04 | PVP | 744 ± 22 (n = 2) (316–582 °C) | 788 ± 22 (n = 2) (316–582 °C) |
| Ce-05 | Stearic acid | 188 ± 12 (n = 3) (264–550 °C) | 213 ± 12 (n = 3) (264–550 °C) |

[1] Corrected values are obtained by subtracting the mass change over the same temperature range for a bare sample of the same size from the same supplier.

### 3.1.4. Dispersion of Particles in Water and Stability in Cell Culture Medium

DLS for suspensions of nanoparticles dispersed using the optimized sonication energy (Table S1) indicate that all samples have Z-average values that are larger than approximately 200 nm, indicating significant levels of agglomeration. In some cases (Ce-06, Ce-07, Ce-08) the PDI values are in excess of 0.4, meaning that the cumulants analysis cannot reliably be used to estimate the particle size. Ce-05 has the largest Z-average and a PDI of 0.22, consistent with strong evidence of aggregation from both TEM and SSA measurements. Ce-05 was dispersed in ethanol for this measurement and when dispersed in water had a size larger than 1000 nm and a PDI over 0.5. This aqueous dispersion was used for cell culture experiments. Several of the cerium oxide nanoparticles exhibit significant agglomeration when diluted from their aqueous dispersions into cell culture medium (Supporting Information, Table S2). Measuring the stability in cell culture medium is important for interpreting the results from the biological assays [31]. Ce-01 through Ce-04 show significant nano fractions that increase in size over a 48 h incubation period; however, between 48 and 72 h Ce-01 and Ce-02 show a decrease in their hydrodynamic diameters, while Ce-03 increases significantly, and Ce-04 increases only modestly. Ce-05 through Ce-08 all were highly agglomerated in cell culture medium when measured by DLS immediately after sample preparation. All eight of the nanoparticles precipitated between measurements and were resuspended by pipetting the suspensions up and down several

times to redisperse the particles. It is not surprising that Ce-05 is highly agglomerated as the stearic acid coated sample is not readily dispersible in water. From these studies it is clear that the mkNano samples lack stability in cell culture medium, as does the N & Am sample, while the samples from USRN (except for the stearic acid coated material) are much better dispersed in water and in cell culture medium and maintain their stability over a 48 h incubation time with only small increases to the hydrodynamic radii. There is no obvious correlation between size and stability here, but there is a direct correlation between manufacture and the quality of the starting aqueous suspension (high z-average or high PDI being unstable) and stability in cell culture suggesting that the method in which these samples are prepared may be important to determining their stability in certain environments and potential bioavailability.

### 3.2. Cytotoxicity of Cerium Oxide Nanoparticles

Cytotoxicity was measured using both the MTT and LDH assays. Over the course of all these assays, white precipitated particles were observed in the wells, and resulted in challenges for measuring the concentration of formazan. For this reason, dissolved formazan in DMSO was transferred to a fresh plate for each well, so that the precipitated particles did not interfere with the absorbance measurements. Some cytotoxicity is seen in most of the 250 μg/mL samples, however, the high levels of precipitated particles may also be interfering with the measurements for these data points. For this reason we discuss the outcomes up to 100 μg/mL as we cannot be certain about the accuracy of the measurements at 250 μg/mL. For Ce-01, Ce-05 and Ce-06 the cytotoxicity at 48 h is observed to be less than at 24 h, however, as it is only observed in this highest concentration, its possible this is an artefact from the materials interfering with the assay as they precipitate. In A549 cells, there is little cytotoxicity observed over 72 h at the concentrations we tested in agreement with other published literature [32]. For the MTT assay, 3 samples (Ce-06, Ce-07 and Ce-08) exhibit some reduction in cell viability below 100 μg/mL, though the % cell viability never dips below 60% for any sample (Figure 4). The lack of cytotoxicity for Ce-05 is likely due to challenges in dispersing the stearic acid coated sample in water resulting lower than expected exposure levels. Ce-06, Ce-07 and Ce-08 were also heavily agglomerated in cell culture medium and the three largest particles sizes by TEM. These samples also exhibit increased cytotoxicity over time. This suggests that larger particles may be more cytotoxic and that high levels of particle agglomeration may be important for the generation of the cytotoxic effects. The results from the MTT assay do not correlate well with results from the LDH assay (Figure 5). The LDH assay measures membrane integrity while the MTT assay measures mitochondrial activity. These are both considered valid measurands for cytotoxicity, but they do not always agree depending on how the material that is being tested impacts the cell. For this assay, the stearic acid coated sample, Ce-05, exhibits the highest level of LDH leakage (Figure 5). Ce-03, which also does not exhibit any significant cytotoxicity in the MTT assay also exhibits LDH leakage at concentrations below 100 μg/mL. There is no obvious correlation with the physical characterization measurements for these results from the LDH assay. This highlights the challenges in selecting assays for general screening using a single assay as even within the same family of nanomaterials, different coatings or even samples from different manufacturers may exhibit different effects and thus affect different cytotoxic pathways. Ce-03 is an uncoated sample and nearly identical to Ce-02 (same supplier and both uncoated and reported to be the same size), yet Ce-02 shows no LDH leakage but somewhat higher cytotoxicity at high doses in the MTT assay.

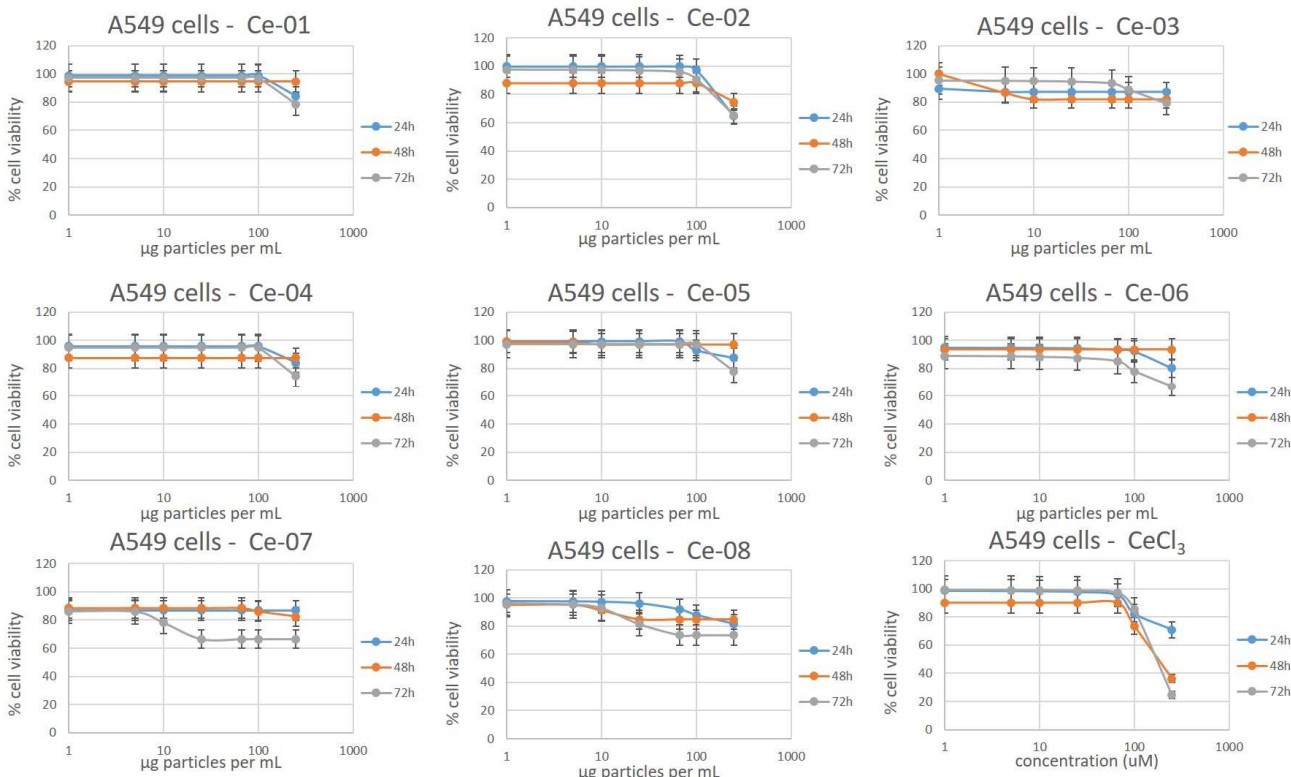

**Figure 4.** MTT data for cerium oxide nanoparticles in A549 cells at 24, 48 and 72 h. Data for an ionic CeCl₃ control under identical dilution conditions is also shown. Raw data was fit to a four variable sigmoidal dose response curve for all samples.

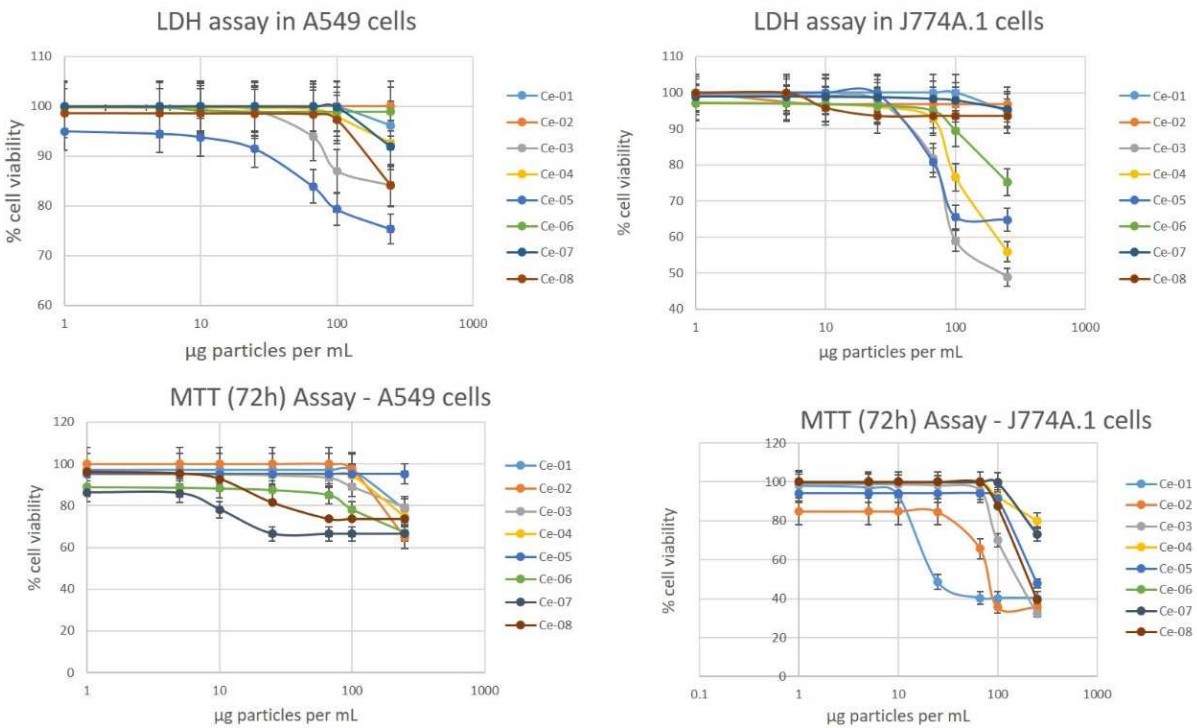

**Figure 5.** Comparison between MTT results at 72 h, and results from the LDH assay also at 72 h in both A549 and J774A.1 cells. Data is fit to a four variable sigmoidal dose –response equation.

Cerium oxide nanoparticles exhibit higher levels of cytotoxicity in the J774A.1 macrophage cell line (Figure 6); however, the observed trends are different than in A549 cells. In J774A.1 cells there is a time dependent cytotoxicity for all 3 of the bare samples from USRN (Ce-01, Ce-02 and Ce-03). These samples were all well dispersed in cell culture medium unlike the samples that exhibited some cytotoxicity in the A549 cells. Unlike in A549 cells, it appears that in J774A.1 cells, agglomerated materials are less cytotoxic than better dispersed ones in general, and larger bare particles are more cytotoxic than smaller or coated materials with Ce-01 being more cytotoxic than Ce-02, which in turn is more cytotoxic than Ce-03. In J774A.1 cells, there is again a difference between the MTT and LDH assay results. The stearic acid coated sample is again cytotoxic in the LDH assay, in fact the 3 samples that exhibit the highest cytotoxicity are the three samples from USRN reported to be 10 nm–bare, PVP coated and stearic acid coated. Ce-06 also shows LDH leakage at higher concentrations, so it is difficult to account for the results of this assay solely on the difference of size, agglomeration or surface coating, though Ce-06 is also a smaller particle; however Ce-02 is about the same diameter and does not show LDH leakage. The results for LDH leakage do not seem to correlate as well with measured physical properties of the particles though there is some indication that in J774A.1 cells that smaller particles may be more likely to cause LDH leakage. It has been shown that for Raw 264.7 cells, LDH leakage is observed for cerium oxide nanorods with larger aspect ratios but not for nanoparticles [18]. The aspect ratio of all the particles in our study are approximately 1.3–1.4, much smaller than in this other study. It is interesting that there is a high level of discrepancy again between the MTT and LDH assay results as well as between cell lines. Further testing is needed to better understand why this is the case for these cerium oxide nanoparticles. While it is not uncommon to have interferences from nanomaterials in cytotoxicity testing, it is curious that, for example, Ce-02 and Ce-03, both small uncoated samples from the same manufacturer of approximately the same size both exhibit similar results for MTT assay in J774A.1 cells, yet vary significantly in the LDH assay.

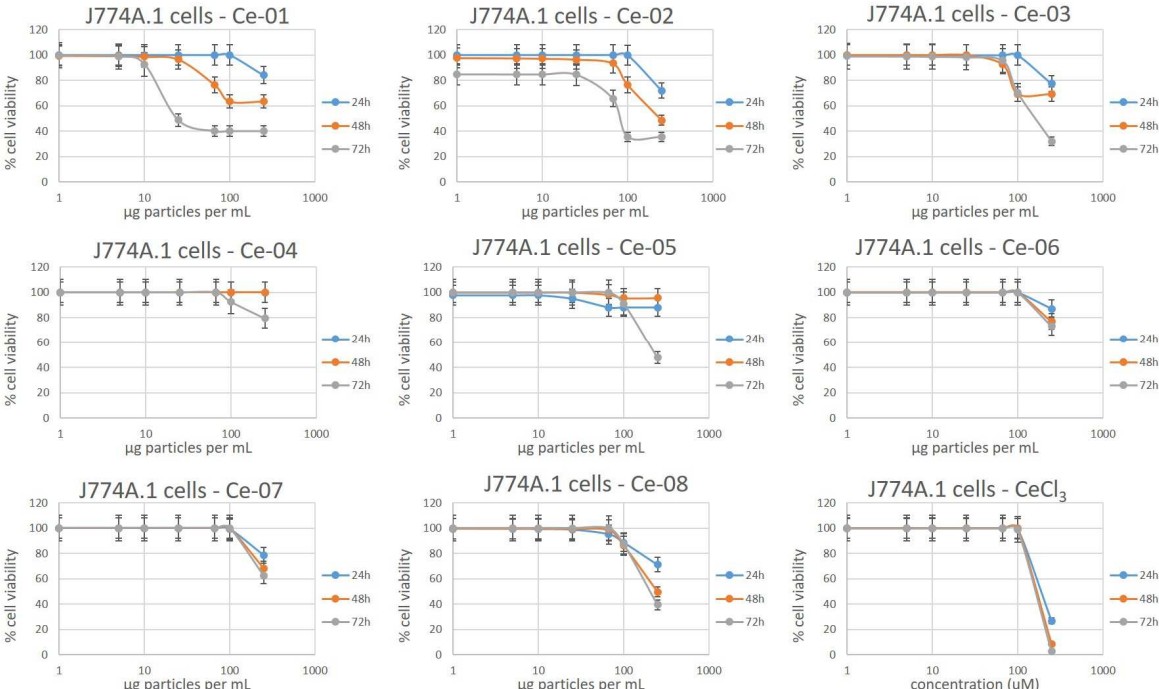

**Figure 6.** MTT data for cerium oxide nanoparticles in J774A.1 cells at 24, 48 and 72 h. Data for an ionic CeCl$_3$ control under identical dilution conditions is also shown. Raw data was fit to a four variable sigmoidal dose response curve for all samples.

### 3.3. Oxidative Stress Measurements of Cells Treated with Cerium Oxide Nanoparticles

We investigated if the cerium oxide nanoparticles contributed to oxidative stress (Figures S4 and S5). In 2006, Lin et al., reported the cytotoxicity of 20 nm cerium oxide particles in A549 cells and found dose dependent toxicity that was consistent with an increase in oxidative stress [21]. The description of the particles in suspension though, differs from our measured results of the commercial particles that were tested. Our higher levels of agglomeration as well as the use of a different cell culture medium may be the reason for our differing results. There is no observed increase in oxidative stress over 24 h for any of the samples, though there is a dose dependent increase in ROS observed for our ionic control, $CeCl_3$. This suggests that the particles are not likely dissolving in cell culture medium over the 24 h time course of the experiment, otherwise a dose dependent result from the dissolved ions would be expected. In general, samples do show a small decrease in oxidative stress compared to controls, but trends are not obvious and the small changes may be a result of lower cell viability compared to the controls. It has recently been reported that small cerium oxide nanoparticles can induce oxidative stress in retinal pigment epithelial cells, so the production of ROS may be cell line dependent [22]. Differences in ROS production and cytotoxicity compared to other published between cell lines suggest that more work is needed to establish read across trends.

In J774A.1 cells, Ce-01 and Ce-03 both show an increase in oxidative stress at higher concentrations, while Ce-04, shows a decrease in oxidative stress at higher doses. This shows some parallels to the MTT results with samples that show an increase in oxidative stress tending to be better dispersed and more cytotoxic while more heavily aggregated samples exhibit no increase in oxidative stress and are less cytotoxic. These trends; however, are not universal as Ce-02 is well dispersed and did not exhibit any significant oxidative stress yet also showed cytotoxicity in the MTT assay. Though it should be noted that for Ce-02 the cytotoxicity occurs at longer timescales than 24 h and so it's possible that ROS production may occur, but at longer exposure times than were included in this study. Ce-04 is well dispersed and shows oxidative stress at lower concentrations but protection against ROS at higher doses. Ce-04 also exhibits significant LDH leakage while not showing any cytotoxicity in the MTT assay making this particular PVP-coated sample unique in its biological activity in this cell line.

### 4. Conclusions

Cerium oxide nanoparticles show tremendous promise in the treatment of several biological disorders where oxidative stress and damage are key drivers such as in neurodegenerative disorders, cancer, bacterial infection and general inflammatory diseases. Our results show that there is often a large discrepancy between reported size and surface area values from manufacturers and those measured in our lab. Using our measurements we were able to elucidate some trends in cytotoxicity, however, these trends were cell line and assay specific. Our results show that the stability and agglomeration of cerium oxidenanoparticles is critically important to their cytotoxicity in A549 cells using the MTT assay, and that in macrophages, larger nanoscale particles were found to be more cytotoxic than smaller ones of similar composition and surface coating. Results for cytotoxicity differed depending on the selected assay with the LDH and MTT assays giving very different outcomes. Testing against a larger number of cell lines might help resolve this discrepancy. For the LDH assay in both cell lines treatment with Ce-03 resulted in LDH leakage whereas the nearly identical particle Ce-02 from the same manufacturer did not, suggesting that maybe the MTT assay is a more reliable assay for measuring cytotoxicity for cerium oxide nanoparticles.

**Supplementary Materials:** The following are available online at https://www.mdpi.com/article/10.3390/ijtm2040039/s1.

**Author Contributions:** Conceptualization, data acquisition, data analysis, writing and editing were performed by M.B., F.K., L.J.J. and D.C.K. X.D. acquired TEM images and A.Z. acquired BET data. All authors have read and agreed to the published version of the manuscript.

**Funding:** Funding has been provided by Health Canada and the National Research Council Canada.

**Institutional Review Board Statement:** Not applicable.

**Informed Consent Statement:** Not applicable.

**Data Availability Statement:** The data that support the findings of this study are available from the corresponding author upon reasonable request.

**Acknowledgments:** M.B. and L.J.J. acknowledge support from a Discovery Grant from the Natural Science and Engineering Research Council.

**Conflicts of Interest:** The authors declare no conflict of interest.

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
