# Peer review of "Characterization of Engineered Cerium Oxide Nanoparticles and Their Effects on Lung and Macrophage Cells"

_2673-8937, doi:10.3390/ijtm2040039_

Round 1
Reviewer 1 Report
dear authors,
the authors characterized 8 cerium oxide nanoparticles using TEM, BET, TGA, and DLS and performed some basic cytotoxicity experiments. The work seems to be well-performed, and while it is not new, it is useful information for those wishing to use the nanoparticles tested. There seem to be some peculiar errors/unclarities) that should be straightened out before the manuscript can be judged suitable for publication.
Major
Line 74: it is assumed that for Ce-05, ethanol was used instead of water throughout the biological experiment (of course diluting in medium) and that this also holds for the respective controls. This is not clear from the manuscript.
Line 88: in my calculation, the final serum concentration is 2.5% , not 7.5%. Line 105: in my calculation, the serum concentration is 5%, not 7.5%. So, the concentrations seem to be different?
Line 254: Ce-06, Ce-07, and Ce-08. ICe-07 and Ce-08 have the largest equivalent diameter distribution, followed by Ce-01 and Ce-06. If Ce-06 is mentioned why not also include Ce-01? Line 260: the reverse: if Ce-01 is mentioned, why not also include Ce-06? This is confusing. Line 380: The same as for line 260.
Figure 4: for Ce-01, Ce-05, and Ce-06, the viability after 24 hour is lower than after 48 hours. Please explain/discuss.
Figure 6: it seems that for Ce-06 the 24 hour-data is missing?
3.3: why not include a Figure?
Lines 476-477: in macrophages, larger nanoscale particles were found to be more cytotoxic than smaller ones of similar composition and coating. Is this true? Not for A549? My confusion also stems from the fact that the results/discussion on cytotoxicity (lines 365-431) is not clear and should be rewritten.
Minor
Line 108: Each day samples were incubated at 37C... I assume this means that they were kept at a constant 37C until the measurements?
Line 115: Please explain sccm
Lines 162-163: The final sentence should be moved to the next paragraph.
Lines 198-199: Why was the fluorescence read before starting the experiment (by adding the cells)? Background measurement?
Lines 376-377: Ce-05 exhibited some cytotoxicity above 100 ug/mL: it was stated before that concentrations >100 ug/mL were not taken into consideration (which is fine).
There are some errors in grammar:
Line 121: at were prepared => were prepared at
Line 259: pf => of
Line 283: C3 => Ce
Line 311: TGA => by TGA
Line 344: PDI => a PDI
Line 361: manufacture => manufacturer
Line 387: meurans => measurands
Line 388: being => that is being
Line 395: effect => affect
Line 452: compared to other published between cell lines??
Author Response
Reviewer 1
dear authors,
the authors characterized 8 cerium oxide nanoparticles using TEM, BET, TGA, and DLS and performed some basic cytotoxicity experiments. The work seems to be well-performed, and while it is not new, it is useful information for those wishing to use the nanoparticles tested. There seem to be some peculiar errors/unclarities) that should be straightened out before the manuscript can be judged suitable for publication.
Major
Line 74: it is assumed that for Ce-05, ethanol was used instead of water throughout the biological experiment (of course diluting in medium) and that this also holds for the respective controls. This is not clear from the manuscript.
Apologies that this was unclear. Although the experiments to check for the optimal sonication energy were done in ethanol, all biological assays were dispersed in water and diluted in cell culture medium. New text has been added after line 74:
Line 88: in my calculation, the final serum concentration is 2.5% , not 7.5%. Line 105: in my calculation, the serum concentration is 5%, not 7.5%. So, the concentrations seem to be different?
The final concentration is 7.5 percent. 100 uL of a 50% dilution (5% serum) and 100 uL of complete medium (10% serum). We have added text to clarify this : (100 µl of 10% and 100 µl of 5% mixed together)
Line 254: Ce-06, Ce-07, and Ce-08. ICe-07 and Ce-08 have the largest equivalent diameter distribution, followed by Ce-01 and Ce-06. If Ce-06 is mentioned why not also include Ce-01? Line 260: the reverse: if Ce-01 is mentioned, why not also include Ce-06? This is confusing. Line 380: The same as for line 260.
We agree that this may have been confusing. The comment (line 54) refers to the distribution width expressed as a ratio of standard deviation/mean equivalent diameter. The values for this ratio are 0.57, 0.43 and 0.43 for Ce-06, -07 and -08. However Ce-01 is only slightly lower (0.37) so the text has been modified. The sentence (originally at line 360) stating that the distribution width is largest for Ce-06 is still accurate but we have now indicated that this is based on the SD/mean ratio.
Figure 4: for Ce-01, Ce-05, and Ce-06, the viability after 24 hour is lower than after 48 hours. Please explain/discuss.
We have added text to discuss this point. As it is only observed in the highest concentration and it has already been noted that there are challenges in correcting for interferences at this concentration we have added the following text: . For Ce-01, Ce-05 and Ce-06 the cytotoxicity at 48 hours is observed to be less than at 24 h, however, as it is only observed in this highest concentration, its possible this is an artefact from the materials interfering with the assay as they precipitate.
Figure 6: it seems that for Ce-06 the 24 hour-data is missing?
The 24 h data is present and can be observed at the highest concentration, however the data is overlayed with later time points and difficult to discern. We have attempted to make the figure higher resolution to make this easier to observe.
3.3: why not include a Figure?
Figures are included in the supporting information.
Lines 476-477: in macrophages, larger nanoscale particles were found to be more cytotoxic than smaller ones of similar composition and coating. Is this true? Not for A549? My confusion also stems from the fact that the results/discussion on cytotoxicity (lines 365-431) is not clear and should be rewritten.
The reviewer has kindly pointed out that this statement was reversed. The larger particle (Ce-01 is the most cytotoxic. The statement was incorrectly referring to the DLS measurements in cell culture medium where Ce-01, despite being a larger particle, has the smallest hydrodynamic diameter in cell culture medium. Several parts of the discussion on the cytotoxicity in J774a.1 cells has been written in order to add clarity and better distinguish which assays show which trends.
Minor
Line 108: Each day samples were incubated at 37C... I assume this means that they were kept at a constant 37C until the measurements?
Response. Yes, that is correct and the text has been modified.
Line 115: Please explain sccm
Response. This refers to standard cm3/minute, which has now been added
Lines 162-163: The final sentence should be moved to the next paragraph.
Response. The information was move to the following paragraph.
Lines 198-199: Why was the fluorescence read before starting the experiment (by adding the cells)? Background measurement?
Lines 376-377: Ce-05 exhibited some cytotoxicity above 100 ug/mL: it was stated before that concentrations >100 ug/mL were not taken into consideration (which is fine).
The reference to Ce-05 has been removed to be consistent with ignoring the highest concentration point
There are some errors in grammar:
Line 121: at were prepared => were prepared at Corrected
Line 259: pf => of Corrected
Line 283: C3 => Ce Corrected
Line 311: TGA => by TGA Left as is, which appears to be correct
Line 344: PDI => a PDI Corrected
Line 361: manufacture => manufacturer Corrected
Line 387: meurans => measurands We did not find this error
Line 388: being => that is being Corrected
Line 395: effect => affect Corrected
Line 452: compared to other published between cell lines => This sentence has been re-written: This shows some parallels to the MTT results with samples that show an increase in oxidative stress tending to be better dispersed and more cytotoxic while more heavily aggregated samples exhibit no increase in oxidative stress and are less cytotoxic.
Reviewer 2 Report
In this work the authors study the cerium oxide nanoparticles interactions with cells to better understand their fate in and impact on biological systems.
The results showed that there is a large discrepancy between reported size and surface area values from manufacturers and those measured by them.
They found some trends in cytotoxicity, correlated to cell line and specific assay.
They focused on the stability and agglomeration of cerium oxide nanoparticles that is critically important to their cytotoxicity in A549 cells using the MTT assay, and that in macrophages, larger nanoscale particles were found to be more cytotoxic than smaller ones of similar composition and surface coating.
For the LDH assay in both cell lines treatment with Ce-03 resulted in LDH leakage whereas the nearly identical particle Ce-02 did not.
The MTT assay resulted to be a more reliable assay for measuring cytotoxicity for cerium oxide nanoparticles.
The only advice is to increase the number of articles in the references.
Author Response
Reviewer 2
In this work the authors study the cerium oxide nanoparticles interactions with cells to better understand their fate in and impact on biological systems.
The results showed that there is a large discrepancy between reported size and surface area values from manufacturers and those measured by them.
They found some trends in cytotoxicity, correlated to cell line and specific assay.
They focused on the stability and agglomeration of cerium oxide nanoparticles that is critically important to their cytotoxicity in A549 cells using the MTT assay, and that in macrophages, larger nanoscale particles were found to be more cytotoxic than smaller ones of similar composition and surface coating.
For the LDH assay in both cell lines treatment with Ce-03 resulted in LDH leakage whereas the nearly identical particle Ce-02 did not.
The MTT assay resulted to be a more reliable assay for measuring cytotoxicity for cerium oxide nanoparticles.
The only advice is to increase the number of articles in the references. 2 references have been added.